# A Hybrid Approach: Dynamic Diagnostic Rules for Sensor Systems in Industry 4.0 Generated by Online Hyperparameter Tuned Random Forest

**Ahlam Mallak \* and Madjid Fathi**

Department of Electrical Engineering and Computer Science, Knowledge-Based Systems and Knowledge Management, University of Siegen, 57076 Siegen, Germany; fathi@informatik.uni-siegen.de

\* Correspondence: Ahlam.mallak@Ymail.com; Tel.: +49-1522-948-8004

**Abstract:** In this work, a hybrid component Fault Detection and Diagnosis (FDD) approach for industrial sensor systems is established and analyzed, to provide a hybrid schema that combines the advantages and eliminates the drawbacks of both model-based and data-driven methods of diagnosis. Moreover, it shines light on a new utilization of Random Forest (RF) together with model-based diagnosis, beyond its ordinary data-driven application. RF is trained and hyperparameter tuned using three-fold cross validation over a random grid of parameters using random search, to finally generate diagnostic graphs as the dynamic, data-driven part of this system. This is followed by translating those graphs into model-based rules in the form of if-else statements, SQL queries or semantic queries such as SPARQL, in order to feed the dynamic rules into a structured model essential for further diagnosis. The RF hyperparameters are consistently updated online using the newly generated sensor data to maintain the dynamicity and accuracy of the generated graphs and rules thereafter. The architecture of the proposed method is demonstrated in a comprehensive manner, and the dynamic rules extraction phase is applied using a case study on condition monitoring of a hydraulic test rig using time-series multivariate sensor readings.

**Keywords:** industry4.0; fault detection; fault diagnosis; random forest; diagnostic graph; distributed diagnosis; model-based; data-driven; hybrid approach; hydraulic test rig

## 1. Introduction

In the last century, a new notion called "Industry 4.0" has emerged, in which the world has witnessed an industrial and technological revolution that caused the rapid spread of complex sensor systems in various domains and applications. These systems are literally everywhere; in aircrafts, transportation like cars and ships, systems such as computers, laptops, smart phones, embedded systems, and in many industrial applications such as factories, chemical reactors, nuclear power plants, and many more countless examples. As long as these complex systems continue to function properly, they play a major role in providing help, comfort and assistance to our daily lives, and they are even considered a necessity to the current structure of modern societies.

A fault in sensor systems within industry 4.0 can be defined as an unexpected event occurring at a certain point in time, which may trigger bigger events or a series of other unexpected events. Isermann and Balle [1] defined faults as an unauthorized, not permitted or disallowed deviation of one or more of the system's parameters, characteristics, behaviors or patterns from the normal or standard state of the system.

Based on the nature of complex industrial sensor systems mostly being non-linear, dynamic and having complex relationships between their components, it is highly complicated to predict faults in such systems. Fault consequences fall in a spectrum that ranges from harmless, ignorable faults to extremely disastrous ones that may lead to major economical and human catastrophes.

Most research suggests that faults in industrial sensor systems can be classified into three main categories; sensor faults, actuator, and component/system faults [2].

- Component faults are a deviation in the system's characterized patterns and behaviors, occurring in one, some or all parts/components of the system, other than the sensors and actuators, i.e., a fault in the cooling unit in an HVAC system, a fault in the transmission stick in an automobile or a fault in the CPU of a computing system.
- Actuator faults are the faults that occur in actuation units, and appear as a partial or complete malfunction of the actuation control. In other words, the actuators could be faulty when they fail to perform the actuation function, i.e., a stuck actuator. A complete fault in actuators can appear as a result of a burning wire, a cut, leakage, breakage or the presence of an actual physical object holding back the actuator, preventing it from controlling the system's behavior.
- Sensor faults are the faults represented by the sensors and their readings. Usually, these faults are noticed when the sensors are producing incorrect readings due to a physical fault in the sensor itself, broken wires or a malfunction in the communication channels between the sensors and the controlling unit, or the change of the sensor's reading could be an indicator (symptom) of a component or system fault.

Complex sensor systems within industry 4.0 contain hundreds of sensors attached in different locations of the system, wired or wireless, stationary or mobile, to continuously measure some key variables of these systems in real time. The data generated from sensors are considered a rich source of information from the analytical perspective, since this type of data has a vast majority of unique patterns and worthwhile characteristics. Moreover, any sudden changes of these sensors' readings, or the appearance of any unexpected patterns that go without notice, can lead to a major risk and serious consequences.

Fault Detection and Diagnosis (FDD) is the process of finding odd, extraordinary or unusual patterns in given data, compared to the patterns it usually forms in the regular or healthy state. These irregular patterns are most commonly called faults, anomalies or outliers [3]. In the last decade, FDD has been an interesting topic for many researchers, applied in a wide range of applicational domains, due to its enormous significance in providing the needed safety, security and reliability in many industrial systems, as well as the vital role it plays in the fast detection of abnormalities and faulty patterns, which is essential in many industrial systems, especially the ones in harsh or highly restricted environments, systems that are prone to malicious attacks, sensor systems that contain fault-prone sensors or where the sensors' readings might be faulty or unusual. As a result, many FDD systems are developed for a specific domain, while others offer a more generic solution.

FDD can be classified into many groups and different types based on the strategy with which the diagnostic process is approached. The most common methodologies are model-based, data-driven and hybrid approaches [4–6].

- In model-based approaches, the diagnosis process follows a reference represented by a model, which can be in different forms and shapes, such as a set of diagnostic rules, a semantic knowledge-base, a heuristic database, a simulation or an actual physical model. In such approaches, the quality of the diagnosis is completely associated with the accuracy of the created model.
- Data-driven approaches might not offer any knowledge of the physical system or a modelled simulation. However, they tend to extract data-driven knowledge by analyzing the system recorded data and applying various methods to find hidden patterns and relationships that describe the unknown system and its behavior. Although these approaches do not provide a good insight into the system and its processes, these approaches are extracted from the data

(unlike model-based methods that are dependent on fixed rules and rigid knowledge), which is considered a dynamic, general-domain solution. The most famous approach related to data-driven FDD is using machine learning in a statistical manner or a black-box fashion. For statistical machine learning, there is a fair share of pattern recognition algorithms that are represented by various classifiers, regressors and clusters, designed to solve numerous problems based on the availability of the class or label of the observations. In the black-box approach, shallow or deep neural networks can be applied to learn various diagnostic models based on the data. Data-driven diagnostic quality is dependent on various factors, which are all related to the data, such as the performance in data generation, data pre-processing, feature selection and/or feature extraction, and finally the machine learning algorithm chosen to solve the diagnostic problem.

- Hybrid-based approaches are the ones created by different approaches from the same group or different groups collaborating all together to finally create a new diagnostic offspring approach, that possesses shared qualities from the original parent approaches, and hopefully is sufficient to eliminate one or more of the parent approaches' drawbacks compared to when applied individually, i.e., a combination of two data-driven approaches to form a new hybrid one, or establishing a bridge between a specific method in data-driven and model-based to finally produce a hybrid offspring of shared qualities.

## 1.1. Challenges and Problem Statement

Model-based approaches to FDD tend to have a handful of disadvantages, such as the lack of dynamicity and generality, since they exhibit static knowledge for a specific domain stored in the model, the lack or absence of an ability to handle sudden or novel fault occurrences (hence they are not pre-stored in the reference model), and the inability to automatically detect, fill or update the system gaps. They also lack credibility in knowledge acquisition, because it is completely dependent on experts' reliability. Finally, they are unable to learn from misdiagnoses and fault occurrences over time.

Although data-driven approaches might offer dynamic and general-domain diagnostic alternatives compared to their model-based counterparts, they tend to have their share of challenges. For example, the dependency on the data in the case of poor data collection, or their tending towards invalid sources, their dependency on possessing certain skills to apply data-driven processing and analysis methods. Furthermore, storing data necessary for learning and testing is resource- and security-expensive, and there are additional expenses related to the needed supplementary hardware purchases and regular maintenance.

According to the mentioned drawbacks of each approach, it is essential to establish a hybrid approach that combines the positive sides of each one and eliminates as many as possible of their limitations.

This work is intended as an extension to the method in [7]. The algorithm in [7] demonstrates a model-based component FDD method based on using diagnostic graphs created by static/constant diagnostic rules extracted from semantic ontology. In other words, the system model represented by the ontology [8] is fed directly with the expert knowledge, and later used to generate diagnostic graphs that link between various symptoms and their faults. The graphs created using the model-based approach alone are lacking the dynamicity and the generality, as they are only applicable to a certain system or model that they were created upon. Thus, a more general and dynamic approach is needed.

Creating dynamic diagnostic graphs using data-driven approaches such as Random Forest (RF) can be beneficial. However, because of their dynamic nature, these models are hard to use in structured or distributed systems without following some guidelines, graphs or clear steps. Furthermore, data-driven approaches require more time and resources to process and store the needed data. Hence the strong necessity of creating a general-domain, dynamic but structured enough algorithm, to guarantee general-domain application and a decrease in the time and resource complexity constraints experienced by online data-driven approaches.

*1.2. Our Contribution*

On one hand, this work demonstrates a unique architecture to deploy RF in FDD beyond its ordinary application as a data-driven methodology. Usually, RF is used as a supervised learning algorithm to classify subjects into various classes. As well as the use of RF for feature selection i.e., feature importance method. Moreover, when certain adaptations are made RF can also be used for unsupervised learning. However, the literature is lacking the use of RF for model-based FDD or hybrid approaches beyond the data-driven combination ones.

On the other hand, the architecture and application of an algorithm to diagnose component faults for industrial sensor systems, in a dynamic and distributed fashion, is introduced, where a hybrid approach between model-based and data-driven methods is established.

Finally, in this work, a development and extension of the work in [7] is proposed, by offering a dynamic and general-domain approach, with the possibility of deployment in distributed systems, represented in generating dynamic diagnostic graphs using RF.

It is worth emphasizing the reasons behind choosing RF to contribute as part of this work, instead of many other machine learning classifiers. RF has been chosen in this work due to its high accuracy for fault classification when applied to real-life datasets, and compared to a wide variety of other classifiers of different functionalities and workflows. This comparison is explained in detail in the Results section. Furthermore, for the creation of diagnostic graphs, it is easier to incorporate a classifier that provides some sort of directed graph structure in its flow, which is supported by the tree structure of RFs.

The rest of the work is organized as follows: Section 2 shows RF-related work for FDD in industry 4.0. In Section 3, some of the literature of RF is introduced. Section 4 provides the materials and methods used in this work. Section 5 demonstrates the system overview and experimental results. Finally, Section 6 represents the discussion and future work.

## 2. Related Work: RF for FDD in Industry 4.0

For the last few decades, RF has been used widely to perform FDD and monitoring applied in various fields and applications, such as in industry 4.0. The literature demonstrates several techniques to apply RF for the purpose of outlier detection, either exclusively or incorporated with other algorithms to form some sort of a hybrid approach aimed at fulfilling an intended research or applicational purpose.

The most common methodology of deploying RF is using it as a classifier. RF is intended to achieve an optimized, supervised and structured resolution for labelled problems, which is proven to given more accurate results compared to many other supervised machine learning algorithms. In [9], RF is compared to numerous classifiers of different functionalities for its ability to overcome two occurring sensor faults in Wireless Sensor Networks (WSNs), which are spike fault and data loss fault. This study represents an elaborated comparison between RF, Support Vector Machine (SVM), Stochastic Gradient Descent (SGD), Multilayer Perceptron (MLP), Convolutional Neural Network (CNN) and Probabilistic Neural Network (PNN), using Detection Accuracy (DA), Matthews Correlation Coefficients (MCC), True Positive Rate (TPR) and F1-score as the comparison criteria that determine the overall rank of each method. As a result, RF is proven to have the highest rank of all the above classifiers in the WSN's sensor fault classification. In addition, another study in [10] showed similar results in proving the superiority of performance of RF in the field of WSN, but this time while detecting four different sensor faults; gain, offset, constant and out of range faults.

In [11], we see another example of using RF in a solo fashion to achieve FDD in industrial sensor systems applied to unmanned aircraft vehicles. This study deployed a brilliant interpretation of RF and feature importance to extract a weighted similarity metric based on the data priority represented by RF. The induced similarity measure is then used to perform FDD.

RF can also be used combined with different approaches, instead of using it directly as a classifier to achieve FDD in industrial systems. Usually, any hybrid approach is developed to optimize the individual forming methods combined, or to establish a customized solution that fulfils additional system goals or requirements.

In [12], a hybrid approach is established to detect faults of rolling bearings, which if left undetected can lead to major consequences in the performance of the rotating machine. This hybrid approach combines the Wavelet Packet Decomposition (WPD) method to extract new enhanced features from the bearing vibration signal provided by *n* number of sensors, using signal-to-noise ratio and Mean Square Error (MSE), followed by the step of mutual, dimensionless index construction, which will be fed to the fault database and contribute as the data necessary to train and test the RF model.

Moreover, another example of a hybrid FDD approach using RF is [13]. This method demonstrates the effect of combining a genetic algorithm and RF to increase the classification accuracy of the FDD process of an induction motor.

In this recent work [14], RF is used in a hybrid fashion with Feedforward Neural Network (FNN) to investigate the relationship(s) among multi-modal signals, extracted from electrochemiluminescence (ECL) sensors located in a smartphone and the concentration of $Ru(bpy)_3^{2+}$ luminophore and its electrochemical data. Establishing such a correlation is essential for building optimized and cheaper diagnostic devices. Understanding the hidden relationships between each modality may lead to creating diagnostic rules, which can be used for FDD in later stages. Thus, this study is included in the applications of RF in FDD-related work.

Beyond the intensive use of RF in industrial sensor systems, RF can be used in a smaller range, for many reasons and purposes exceeding the industry. One of the common applications of RF is in the medical field, using sensing modalities. In [15], a recent study has shown the application of RF in reducing fallacious clinical alarms, i.e., the arrhythmia alarms, in cases of false arrhythmia alarm occurrences, that may lead to elevation in patient and staff stress level, as well as causing unnecessary pressure on the intensive care staff. According to the study, the application of RF in discerning the true from the faulty calls has significantly reduced the amount of false calls, concerning five main types of arrhythmia.

## 3. RF Literature Review

Random forests (RF), also known as Random Decision Forests (RDF), is a machine learning algorithm of an ensemble nature that is designed for classification and regression purposes. RF is an ensemble learning algorithm based on decision trees methodology, which is intended to optimize decision trees and resolve their tendency to form overfitting patterns over the training dataset.

In order to understand the mathematics behind RF, it is highly recommended to go through the explanation of decision trees and how they work in the first place.

### 3.1. Decision Tree

Decision trees in data mining are a commonly used supervised technique to solve classification and regression problems, where a set of observations and their labels or classes are already known and used to make various predictions [16]. In data mining, decision tree algorithms are divided into two main types: classification and regression trees. In 1984, Breiman et al. [17] combined the two types together under the same category, using the term Classification and Regression Tree (CART).

Decision trees are called this way because they are visualized in a tree structure, which is created by recursively splitting the training dataset from top to bottom, forming the first level node of depth zero called the "root", followed by going down the tree, forming greater depths and continuously splitting into successor children nodes. The splitting process is determined using different rules that determine the impurity of a certain node, which influences the selection of the splitting criteria [18].

Table 1 represents some of the most common node impurity criteria based on the tree type, linked to their scientific formula, and a brief description of the splitting mechanism.

In this work, we will only focus on classification trees. Thus, only classification trees' splitting criteria are discussed and deeply explained. For more information check [19].

**Table 1.** Node Impurity Splitting Criteria.

| Tree Type | Criterion | Mathematical Formula | Description |
|---|---|---|---|
| Classification Tree | Gini Impurity Entropy | $\sum_{i=1}^{C} f_i(1 - f_i)$ $\sum_{i=1}^{C} -f_i \log(f_i)$ | $f_i$ is the frequency of the class $i$. Where $C$ is the total number of classes/labels |

Gini Impurity is one of the most common metrics used to determine the best split for classification trees. Gini impurity works by finding the probability for each class/label in which they are incorrectly classified. In other words, Gini impurity is the probability of the falsely classified subjects, with respect to a randomly chosen split point based on the original distribution of the dataset. During the decision tree training process, the best split is represented by the split that maximises the Gini gain, where the Gini gain is calculated by subtracting weighted impurities of the left and right branches of the chosen split from the original impurity existing in the whole dataset, before randomly choosing the splitting point.

### 3.2. From Bootstrap Aggregation (Bagging) To RF

Furthermore, there are some algorithms classified under the ensemble learning category that allow the possibility of creating multiple different trees over the same dataset, to contribute to minimizing the over-fitting problem decision trees usually suffer from, especially when the sample size provided is relatively small. The two main types of such ensemble methods are boosted ensemble trees [20,21] and bootstrap aggregated or bagged trees [20,22].

Boosted trees are a sequential type of ensemble decision trees, wherein the optimal shape of the tree is established incrementally by adjusting the tree continuously based on the arrival of new instances. The most famous boosted trees algorithm is AdaBoost method.

Bootstrap aggregated or bagged trees are a parallel type of ensemble decision trees that generate multiple numbers of decision trees concurrently, by resampling the training dataset with replacements. The final prediction for such methods is made by voting the results of the created trees altogether. What is worth mentioning is that random forests are an example of a Bootstrap aggregating method optimizing the traditional decision trees methodology [22].

Random forests are an optimization algorithm of decision trees, under the ensemble learning sub-category, intended to perform different tasks such as classification, regression, and many others. The core of this algorithm relies on creating a multitude of parallel decision trees based on dividing the feature space each time and deploying the chosen sub-space to form the tree of choice. The prediction decision of random forests is made by the majority vote of all the separately created trees. Random forests started as the "Stochastic discrimination" approach created by Eugene Kleinberg [8], which was inspired by the formula created by Tin Kam Ho [23] in order to deploy the understanding of random subspaces and how to use them in a practical approach. Recently, the random forest algorithm has been trademarked by Leo Breiman and Adele Cutler, and has been owned by Minitab, Inc, since 2019 [24]. The registered algorithm represents an extension of the formula introduced by Ho [23], and the "Bagging" idea created by Breiman [17,22].

The bagging algorithm's whole idea depends on randomly choosing a subset of the original training set with placement, performing an *S* number of classification or regression tasks, and finally making the overall decision of the performed task using all the learners created.

Generally, the trees created by the bagging algorithm alone tend to be highly correlated and, in most cases, the same tree is being generated multiple of times, due to simply training multiple trees over the same dataset with placements that can easily generate high correlations between the formed estimators. The best way to introduce some sort of de-correlation between the trained trees is by feeding the algorithms different datasets. A new dataset can be formed from the original dataset by using the random subspace algorithm [25] to not only randomly choose the data points, but also concurrently

pick randomly a feature from the feature space, to act as a new splitting point. Random forests use the random subspace method to de-correlate the trees formed using the bagging method alone.

The random subspace algorithm is highly similar to bootstrap aggregation in many ways. The only difference is that in random subspace, the features are the subject of bagging, and they are considered the "Predictors" or "Random variables" that would be sampled with replacement to create predictions for each learner. Thus, random subspace is also known as attribute bagging [26] or feature bagging. The random forest approach is a combination of bootstrap aggregation, to sample the training dataset, and the random subspace algorithm, which is necessary to sample the features, so as to create splitting points that result in generating multiple estimators with high levels of distinction and accuracy. The Pseudo code (Algorithm 1) below shows a detailed explanation of random forests.

---

**Algorithm 1** Random Forests

---

1: Given a training labeled dataset
$$(X, Y) = \{(x_1, y_1), (x_2, y_2), (x_3, y_3), \dots (x_n, y_n)\}$$
where $X$ is a set of data and $Y$ is its corresponding response or label
2: For S number of times: Randomly choose N number of random samples with replacement from $(X, Y)$ and M number of features.
for s = 1, 2, ... S:
(a) (Bootstrap Aggregation):
With replacement randomly choose N training samples called Xs, $Y_S$ from (X,Y)
(b) (Random Subspace):
Randomly choose M number of features from the feature space
(c) Choose the best split among the features randomly selected in (b)
(d) Grow the random forest tree ($T_S$) to the data chosen in (a) using the best split
3: Ensemble algorithm output in the form of S number of trees $\{T_s\}_1^S$
4: To make predictions of a new test point x
- Regression Predictions:
The average of all predictions from the S experiment $\hat{f}$ is:
$\hat{f} = \frac{1}{S} \sum\limits_{s=1}^{S} f_s(x)$
- Classification Predictions:
Use majority voting to find the predicted class.

---

## 4. Materials and Methods

*Condition Monitoring of Hydraulic Test Rig Data Set*

This dataset [27] represents real measurements of multivariate, time-series sensors, placed in a hydraulic test rig. The purpose of the data collection is to monitor and assess the hydraulic system's health condition.

The outcome of this experiment yielded a success in collecting sensor data of various system health degrees from different components of the hydraulic system, such as the cooler, valve, pump and accumulator.

The system consists of six pressure sensors, four temperature sensors, two volume sensors and one vibration sensor, which all possess a constant cycle of 60 s. Each cycle the sensors are collected, while the conditions of the four main hydraulic components, cooler, valve, pump and accumulator, are monitored and observed. The component health ranges from completely healthy to totally damaged, and each condition degree is decoded into a numerical value to facilitate the application of statistical and data mining approaches.

This dataset has been used by many researchers to perform sensor fault monitoring, i.e., constant, shift, bias and peak [2]. Moreover, this dataset is also beneficial in performing component or system

FDD, such as the application researched in [28,29]. It is also applicable in creating and testing feature extraction and selection algorithms [30].

## 5. Results

### 5.1. System Model Overview

The proposed system scenario represents a possible method of merging random forest as a data-driven approach to industrial FDD and model-based FDD approaches into a final hybrid approach, which possesses the powerful features of both approaches. This technique eliminates the main drawbacks of each approach individually, such as the lack of dynamicity and responses to sudden occurrences, in the case of traditional model-based FDD. Additionally, it provides validated, accurate and dynamic diagnostic rules that contribute massively in reducing the required diagnostic time and computational resources, compared to their online data-driven counterparts. Figure 1 shows the two main diagnosis phases used in this research, data-driven and model-based, and how these two methods are combined into a new improved approach.

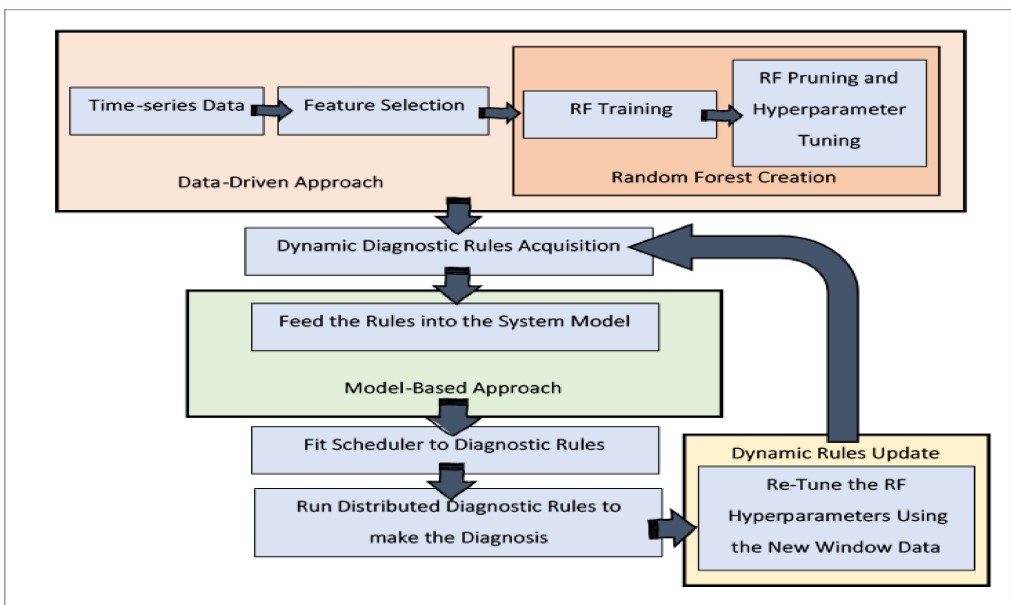

**Figure 1.** Hybrid-based Fault Detection and Diagnosis System Overview.

The following is a comprehensive explanation of each phase:

- Data-Driven Phase:

This phase consists of multiple internal steps essential to learning the best possible dynamic diagnostic rules using a random forest algorithm. Below, each step is discussed in an elaborate manner.

- Multivariate Time-Series Dataset

In this work, the dataset used is a multivariate, time-series dataset, of six pressure sensors, four temperature sensors, two volume sensors and one vibration sensor, which all possess a constant cycle of 60 s, placed in a hydraulic test rig to monitor its condition over time. For more details about this dataset and its previous applications, please refer the data collection and generation section in this work.

- Feature Selection

Complex industrial sensor systems often have hundreds or even thousands of sensors connected, simultaneously transmitting sensor readings data crucial to monitor and control those systems, each of

which is considered a feature for analysis and model training. Thus, creating diagnostic models that only include valuable features is a necessity.

Implementing a model with fewer but more meaningful features has a significant impact on the overall system. First, the diagnostic model becomes simpler to analyze and interpret when fewer elements are included. Second, by eliminating some features of the dataset, the data would be less scattered, hence fewervariants, which can lead to reducing overfitting. Finally, the main reason behind feature selection is to generally reduce the time and computational costs required to train the model.

In practice, the RF algorithm can be applied to carry out feature selection as well, simply because the features are implicitly ranked based on their impurity during the formation of each decision-tree that create the forest. In other words, when top-down traversing a tree in RF, the nodes toward the top happen to have the largest impurity metric reduction, also known as Gini Impurity, compared to the nodes at the bottom. Thus, by determining a particular impurity decrease threshold, it is possible to prune the tree below this tolerance, in order to establish a subset of the most fitting or important features.

The data-driven FDD method implemented in this work is RF. Its intention is reducing the computational cost as much as possible, and RF is also used to perform feature selection using what is known as feature importance or permutation importance [22,31], since Gini impurity calculations are already measured during the RF training process, and only a slight bit of additional computation is required to complete the feature selection process.

In the context of RF, the Mean Decrease Accuracy (MDA), or permutation importance or feature importance, of a variable $X_n$ in predicting $Y$ of classes is computed by the summation of the Gini impurities of $X_n$ for all the nodes $d$ where $X_n$ is present and used, followed by the mean of the impurity decrease metric of all the trees $D$ in the forest. The following equation comprehends the concept of feature importance using RF.

$$Importance(X_n) = \frac{1}{no.Trees} \sum_{no.Trees} \sum_{d \in D : v(s_d) = X_n} Gini\_Impurity(X_n) , \qquad (1)$$

where $X_n$ is the feature of interest, and $v(s_d)$ is the feature/variable used to split $s_d$.

The most popular implementation of feature importance provided by RF is the Python library Scikit-learn, where a pre-defined function, feature_importances_, is directly executed given the learned RF model. However, a team of data scientists at the University of San Francisco pointed out some bugs associated with this function, and implemented an alternative to generate more accurate feature importance results in [32].

- Hyper-Parameter Optimization

The foremost goal of any machine learning algorithm is to minimize the expected loss as much as possible. To achieve this, it is necessary to deploy some optimization equations to select the optimal values for some, or all, of the hyperparameters of the machine learning algorithm of interest.

The RF algorithm has plenty of hyperparameters. On the one hand, some are implemented on the overall forest level, such as the number of subjects randomly drawn from the dataset to form each tree, the choice of with or without replacement regarding the samples selection, and most importantly, the number of trees in the RF. On the other hand, some hyperparameters are on the tree level, which controls the shape of each tree in RF, i.e., the number of features drawn for each split, the selection of splitting rules, the depth of each tree, and many others. These parameters are typically selected by the user. Consequently, creating a method to efficiently select these hyperparameters can influence the performance versus the cost of RF significantly. In addition, the recent research done in [33] emphasizes the significance of hyperparameter optimization specifically for RF parameters, as well as providing deep comparisons between numerous tuning and optimization mechanisms and software.

One of the key tuning strategies for RF is using searching algorithms to look for optimal parameters in a pool or grid of selected ones. Search techniques differ in their way of pool or grid creation, based

on the mechanism applied to choose the successful candidates forming the bag of options. Some searching strategies use all the possibilities available as candidates to be exhaustively investigated, one by one, to select the optimal choice, as in a grid search algorithm. However, in random search, the bag of candidates is drawn randomly from the overall existing possibilities, which is not only a precious asset for reducing the search complexity, but also studies have proved that random search produces better accuracy scores for parameter optimization, compared to grid search [34].

Random search refers to a group of searching algorithms that rely on randomness or pseudo-random generators as the core of their function. This method is also known as a Monte Carlo, a stochastic or a metaheuristic algorithm. Random search is beneficial in various global optimization problems, structured or ill-structured over discrete or continuous variables [35].

Below is the pseudo code describing the workflow of a typical random search algorithm (Algorithm 2).

---

**Algorithm 2** Random Search Algorithm

---

Let *RF* is the cost function to be optimized or minimized. *C* is a candidate solution in the search-space $R^n$.
1: Select termination condition *TC*, i.e., specific fitness measure achieved, or max number of iterations reached, and so on.
2: Initialize C. *C = Random position* $\in R^n$
3: *For TC*
   (a) Randomly choose another position $C_{new}$ from the radius surrounding C (the radius of the hypersphere surrounding *C)*
   (b) if $RF(C_{new}) < RF(C)$ then
$$C = C_{new}$$

---

In practice, the Scikit-learn library for Python machine learning provides a method, RandomizedSearchCV, which can be provoked by creating a range for each hyperparameter's subject of optimization. By using the RandomizedSearchCV method over the predefined range, random search is performed to randomly select a candidate grid of possibilities within the range, then the K-fold cross validation technique is applied over the created grid. For additional examples of this method, refer to [36].

- Model-based Phase

This phase represents a clear model of the system, whether it is an actual physical model, a simulation, a knowledge-base semantically connecting the system component together, or a relational database. Based on the system model's nature, the extracted, nested, conditional rules from the random forest are transmitted to a suitable form, i.e., in knowledge-based systems such as ontologies the rules are converted into SPARQL semantic queries [37], regular SQL queries in case the system model is represented by a relational database, or in a simpler fashion the direct application of the extracted rules as a small conditional code that can be executed every diagnostic window to perform the diagnosis. This phase is crucial to minimizing the online diagnostic time and computational power needed, compared to provoking the testing RF algorithm over and over for each sliding window. Moreover, it provides the possibility of introducing online distributed mechanisms given the rules and the graphs creating them.

- Dynamic Rules Update Phase

In this phase, the new time-series data generated by the system for a certain amount of sliding windows are stored and used to update the originally created RF, by performing the hyperparameter tuning again to find out if any alteration of the RF parameters could reduce the size of the overall RF and increase the accuracy at the same time. The new updates selection or rejection decision is highly dependent on the accuracy of the newly tuned RF.

## 5.2. Experimental Results

In this experiment, RF is used, following the steps in the data-driven flowchart in Figure 1, to generate dynamic diagnostic rules to diagnose and monitor the health of a hydraulic test rig. Provided in the dataset, each component condition ranges between full efficiency, reduced efficiency and close to total failure. In this experiment, for the sake of simplicity, the healthy state is represented by the full efficient cycles, and the failures are represented by the cycles where the component is close to failure, while the partial failure state is excluded. Based on the previous fault description, there are four types of total failure in four different components to monitor: cooler total failure state, valve close to total failure, internal pump has a severe leakage and hydraulic accumulator close to total failure. Table 2 explains the definition of each fault chosen for this experiment and some example cycles that contains each fault.

**Table 2.** Hydraulic Test Rig Chosen Faults and Their Full Description.

| Status | Status Description | Example Cycle No. |
|---|---|---|
| Healthy | All components are healthy and in full efficiency mode. | 1788, 1789,1790 |
| Cooler Fault | Cooler has a total failure, and the rest of the components are fully efficient. | 1,2,3 |
| Valve Fault | Valve close to total failure, and the rest of the components are fully efficient. | 1759, 1760, 1761 |
| Pump Fault | Internal pump has severe leakage, and the rest of the components are fully efficient. | 1675, 1706, 1707 |
| Accumulator Fault | Hydraulic accumulator close to total failure, and the rest of the components are fully efficient. | 1465, 1466, 1467 |

The hydraulic system described in this experiment contains 11 sensor readings from three types of sensors located in different components of the hydraulic test rig: 6 pressure sensors, labelled PS1, PS2 up to PS6, 4 temperature sensors TS1-TS4, and finally 1 vibrational sensor labelled VS1. The readings of all the 11 sensors from various cycles, containing the five different statuses shown in the table above, are collected in one labelled dataset of 11 features necessary to perform RF training and analysis.

As mentioned earlier, the selection of RF as the classification method in this work is done after carefully comparing the results of RF with respect to other famous classifiers, such as Logistic Regression (LR), Linear Discriminant Analysis (LDA), K-Nearest Neighbour (KNN), regular decision tree (CART), Naïve Bayesian (NB) and Support Vector Machines (SVM). The supervised machine learning methods shown above along with RF are used to perform a multi-class classification task, to classify the hydraulic test rig faults described in Table 2. The following table, Table 3, shows the classification results after performing multi-class classification using different classifiers. It is demonstrated clearly that CART and RF have elevated accuracy compared to the rest of the approaches. However, RF is an optimization of CART, which overcomes its tendency to form overfitted relationships with the training dataset.

**Table 3.** RF Accuracy Results Comparison to Some Other Classifiers For Hydraulic Test Rig Fault Classification.

| Method | LR | LDA | KNN | CART | NB | SVM | RF |
|---|---|---|---|---|---|---|---|
| Accuracy | 0.780469 | 0.778646 | 0.932031 | **0.989844** | 0.704167 | 0.923177 | **0.985198** |

The non-zero feature importance method is used to neglect the features with less impact of the learning process, by concentrating only on the features that contribute more to the model's accuracy. The table below shows the importance of each sensor to the RF model calculated using Equation (1).

Table 4 shows the calculated importance of each one of the 11 sensors. There are a variety of options by which these importance values are analyzed and evaluated to achieve feature selection. One can pick the highest importance feature alone to represent all the features, or the highest three, highest six or just the non-zero ones to represent the whole pack. However, the most convincing approach is testing all the possibilities and making a logical accuracy versus complexity trade off. For each selected feature(s) scenario, the RF accuracies and the time complexity given $O(T \log n)$ equation are calculated, where $T$ is the number of trees in the RF and $n$ is the size of the input data used for training, assuming that the number of trees, $T$, is constant for all the feature trials. As such, the time complexity is a factor of input data size, represented by the number of features included without sacrificing much or any of the model accuracy.

**Table 4.** Feature Importance Calculated for Each Sensor Feature.

| Sensor Label | PS1 | PS2 | PS3 | PS4 | PS5 | PS6 | TS1 | TS2 | TS3 | TS4 | VS1 |
|---|---|---|---|---|---|---|---|---|---|---|---|
| Importance × 100 (%) | 0 | 0 | 0 | 29.69 | 16.04 | 14.12 | 6.45 | 8.98 | 8.92 | 9.51 | 2.88 |

In Table 5, four different RF model training experiments were conducted to find out the best number of features required to train an RF on 100 decision trees. In the first trial, the most important feature, PS4, is used alone to train the RF model. The second trial used the top three most important features, PS4, PS5 and PS6. The third trial applied the highest six features. Finally, only the non-zero features were selected to train the RF model. For all the above four experiments, the random forest had fixed hyperparameters which were randomly chosen: 100 trees and the maximum depth of 5. Furthermore, a 10-fold cross validation technique is used to compute each trial's accuracy.

**Table 5.** RF Accuracies Using Different Features Based on Their Importance.

| Trial Description | No. Features | RF Accuracy |
|---|---|---|
| No feature selection | 11 | 0.985 |
| Highest Feature | 1 | 0.707 |
| Highest Three | 3 | 0.977 |
| Highest Six | 6 | 0.986 |
| Non-zero importance | 8 | 0.981 |

For this experiment, the highest six features are used for the training process since these features provided the best accuracy among all trials and showed lower time and space accuracy compared to the training using 11 and 8 features, respectively.

The following figure, Figure 2, shows how the time complexity $O(T \log n)$ and space complexity $O(n)$ for the RF are directly proportional to the number of features used. It is crucial to emphasize that the amount of accuracy sacrificed, and the added complexity tolerances, are totally dependent on the system used and one's preferences, i.e., some other researchers would use the highest three features with 0.977 accuracy, if they are willing to lose more accuracy as the cost for the dramatic drop in both time and space complexities.

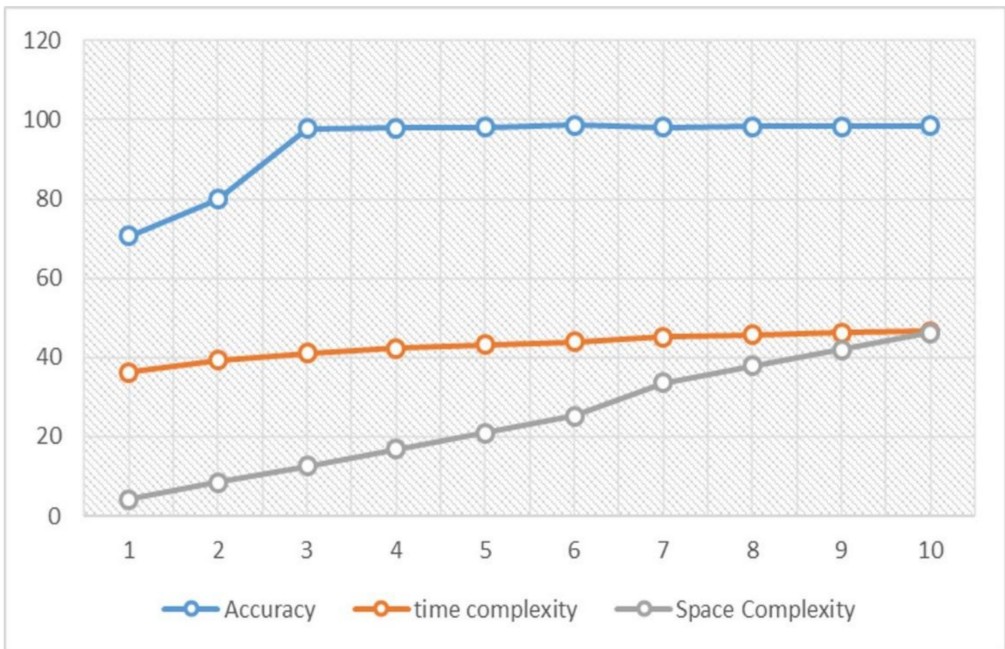

**Figure 2.** RF Accuracy, Time and Space Complexities with Respect to Number of Features.

The next step is tuning the hyperparameters of the RF applied on the dimensionally reduced dataset of the selected six features, PS4, PS5, PS6, TS2, TS3 and TS4. The hyperparameters subject to tuning in this experiment are the number of trees in the RF, the maximum depth of the tree, the minimum number of samples required to split a node and the minimum number of samples required to form a leaf node. As the purpose behind RF creation in this research is to establish a set of base rules for fault diagnosis, the main hyperparameter of focus is the number of trees, to ensure lessening the complexity as much as possible, as well as minimizing each tree's depth if possible.

A random grid of hyperparameters is created by applying a random search over a pre-defined range for each parameter separately, i.e., the number of trees is pre-defined to range between 1 and 1000, and only 100 possibilities are selected from the range to form the random grid for this hyperparameter, followed by RF training using one of the randomly selected pair of features at a time. The selection is validated using three-fold cross validation to calculate the accuracy of the RF model over a particular set of hyperparameters. A set of 100 randomly selected parameters are used to create the grid, which means 300 RF model trainings have been successfully executed, considering the three-fold cross validation over the set of 100 possibilities in the grid. Finally, the set of hyperparameters with the highest accuracy when applying three-fold cross validation is the one selected to generate the diagnostic rules.

The RF model trained after applying feature selection with randomly chosen parameters yielded an accuracy of 0.9865 using 100 decision trees forming the RF, with a maximum depth of five. However, the best hyperparameter tuned using cross validation over the random search grid improved the accuracy by 0.32 to reach 0.9865. Additionally, using only 49 trees in total instead of 100 over the same depth has dramatically decreased the complexity and size of the generated RF rules, while increasing the accuracy and speed of the diagnosis.

The best hyperparameters selected from the grid have 49 estimators, a minimum sample split of two, a minimum leaf samples of one and maximum depth of 83. It is worth mentioning that the accuracy of the RF using all the best hyperparameters increased the accuracy to 0.99, but this slight rise in the accuracy is not easily justified, especially when it is compared to the massive increase in the time and size of each tree in the RF, due to the large increase in the maximum depth.

Figure 3 shows one of the decision trees in the RF after feature selection and hyperparameter tuning.

Each tree in the RF can be translated into a set of nested if-else statements of rules. Moreover, the formed dynamic rules from the RF can be fed inside various system models to generate a hybrid

approach out of the data-driven and the model-based ones. The dynamic rules extracted from the RF can be used as they are, and can be converted to SQL queries if the model is a relational database or SPARQL queries if the system model is represented by a semantic knowledge-base, such as ontologies.

The diagnostic rules can be extracted from the RF dynamically, using a few lines of code in Python language. For the sake of simplicity, Figure 4 shows the tree in Figure 3, pruned in a way that only the positive part of the condition after the root node is remaining, connected to a series of nested statements showing how this part of the tree is translated into clear dynamic rules.

The work in [7] provided a graph-based FDD system for industrial systems using a model-based approach, based on creating a knowledge-base of the system under diagnosis, such as ontologies. Followed by manually feeding a set of static diagnostic rules created by the system expert, into the ontology, in a way that forms a causation relationship between the system sensors and the faults they lead to. In this work, we propose creating dynamic rules using RF, extracting these rules and feeding them into the ontology instead of the expert rules that are static, unreliable and unverifiable. Furthermore, the extracted diagnostic rules using RF can be applied in a variety of forms to fit the model expressing the system.

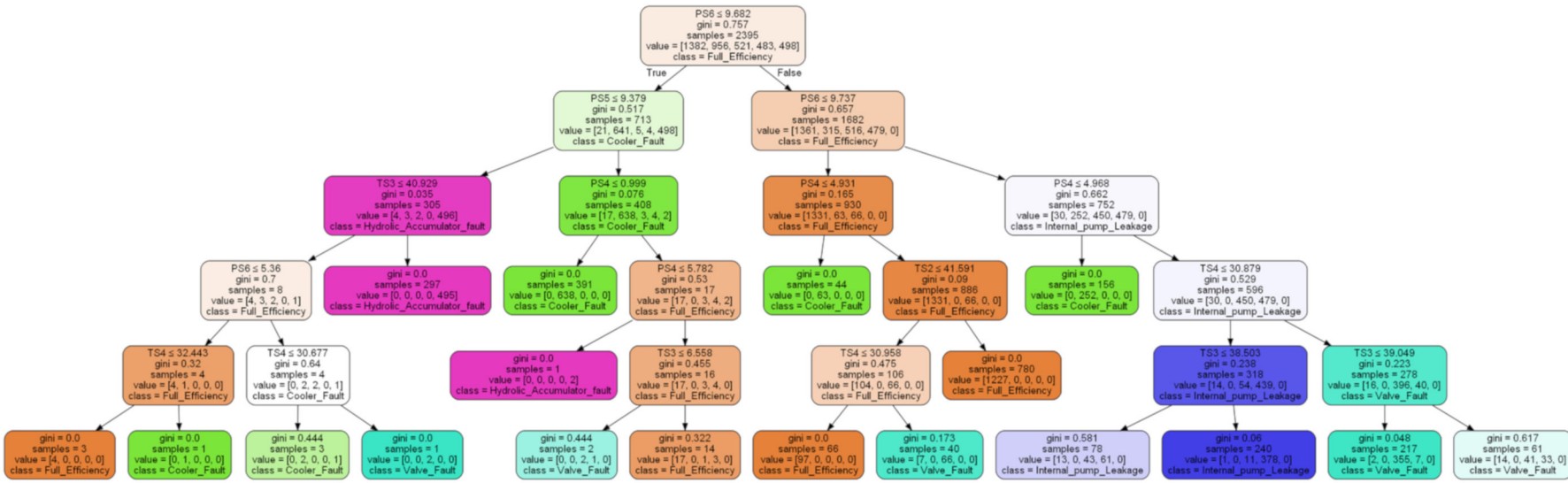

**Figure 3.** One of The Decision Trees in The RF After Feature Selection and Hyperparameter Tuning.

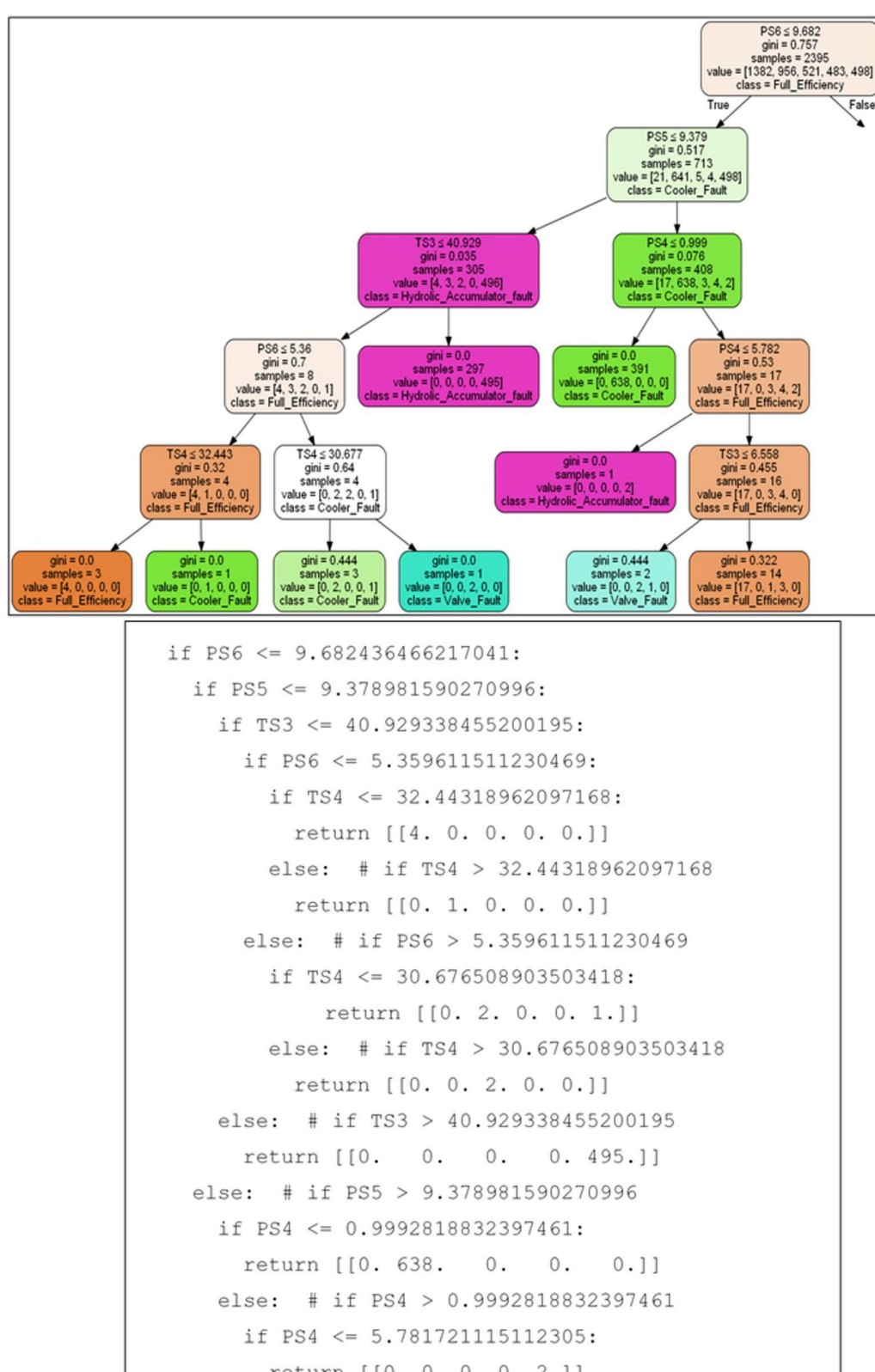

**Figure 4.** Diagnostic Rules Extraction from Parts of A Tree in The RF.

Figure 5, showcases the extracted rules from the optimized, hyperparameter tuned RF, and how these rules are transformed into various forms and types to match the nature of the system model. As mentioned before in this chapter, the diagnostic rules can be translated into SQL queries in cases

wherein a relational database is the system model, or SPARQL queries if a semantic knowledge-base, such as ontologies, are used to represent the system. The rules extracted from each tree may be scheduled separately, or all together with the trees forming the RF.

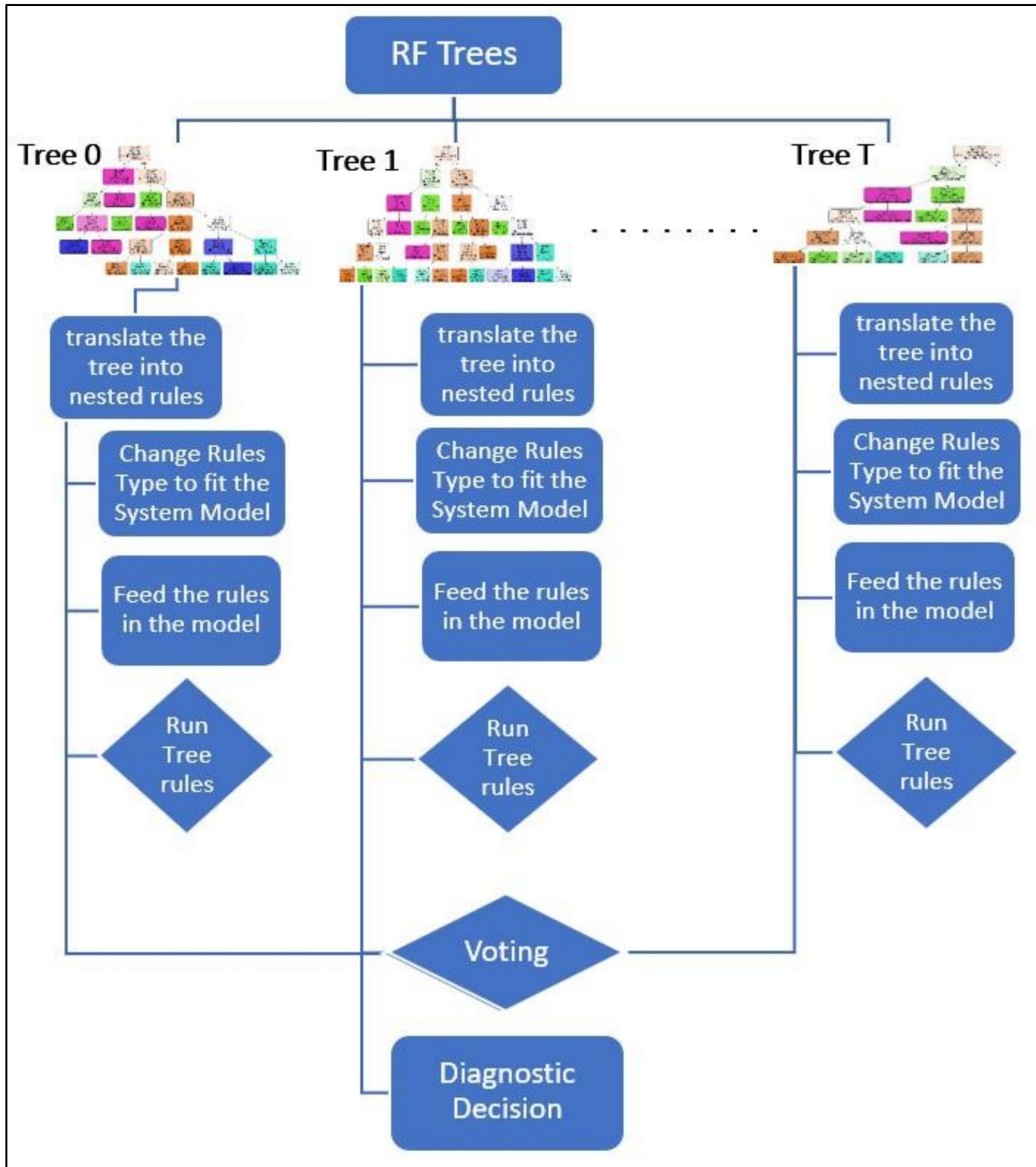

**Figure 5.** A Hybrid RF Approach Between Data-driven and Model-based Approaches.

## 6. Conclusions and Future Work

In this work, the architecture of a hybrid FDD method, between model-based and data-driven approaches to achieve FDD for component faults, is introduced. In this hybrid method, the data-driven part is represented by an optimized and hyperparameter tuned RF, in order to generate dynamic, diagnostic graphs that are later converted into a set of diagnostic rules and fed into a pre-defined system diagnostic model, acting as the model-based part of the proposed FDD system. The proposed approach provides a dynamic solution, unlike other model-based FDD approaches. Additionally, there is the option to apply distributed computing to the diagnostic graphs and extracted rules, which can reduce time and resource complexity compared to traditional data-driven approaches. Moreover, the proposed method introduces a new methodology to approach RF in a model-based fashion, beyond its exclusive, ordinary application as a data-driven approach.

The data-driven part of this system is experimentally applied and analyzed using multivariate time-series sensor data collected from an actual hydraulic test rig. The applied data-driven part includes the RF creation, RF feature selection using non-zero feature importance, and RF pruning and hyperparameter tuning using three-fold cross validation on a grid of variables, selected using random search. Furthermore, the diagnostic rules in the form of nested if-else statements are practically extracted from RF as the diagnostic graph of this approach. The extracted rules can be converted into various forms and shapes depending on the nature of the system model that is the subject of integration.

This work has successfully provided an extension and development of the model-based FDD approach introduced in [7], where the previous system is domain-specific, and can only be applied to the model described in the ontology. It also contains rigid knowledge preserved in the ontology as a set of semantic rules, but later translated into static, domain-specific, application-specific set of diagnostic rules with constrained reliability to the system's expert. On the other hand, our proposed method provided dynamic and reliable diagnostic rules, with the combined advantages of both the data-driven and model-based approaches.

The proposed FDD system offers a vast amount of advantages and new insights. However, there is always room for improvements. Thus, some additional work and further modifications of the proposed system can be applied in the future. In this work, the traditional RF algorithm is applied to serve as the dynamic rule generator in both the offline learning and the online update stage, using the newly arrived sensor reading. Instead, considering online RF methods in the first place, such as Mondrian forests [38] or online incremental RF (described in [39]), may reduce the training and update time. Moreover, in future work a full application of this approach will be introduced and examined in a practical study, where the extracted rules are converted into SPARQL queries to fit the ontology designed of the chosen system. Furthermore, a proper scheduler will be chosen to demonstrate the possibility of distributed computing using the extracted diagnostic rules at run-time.

**Author Contributions:** Conceptualization, A.M. and M.F.; methodology, A.M.; software, A.M.; validation, M.F.; formal analysis, A.M.; investigation, A.M.; resources, M.F.; data curation, A.M.; writing—original draft preparation, A.M.; writing—review and editing, M.F.; visualization, A.M.; supervision, M.F.; project administration, M.F.; funding acquisition, M.F. All authors have read and agreed to the published version of the manuscript.

**Funding:** This research was funded by the DFG research grants LO748/11-1 and OB384/5-1.

**Conflicts of Interest:** The authors declare no conflict of interest.

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
