# Peer review of "A Hybrid Approach: Dynamic Diagnostic Rules for Sensor Systems in Industry 4.0 Generated by Online Hyperparameter Tuned Random Forest"

_sci, doi:10.3390/sci2040075_

Round 1
Reviewer 1 Report
In this work, a hybrid component Fault Detection and Diagnosis (FDD) approach for industrial sensor systems is established and analyzed, to provide a hybrid schema that combines the advantages and eliminates the drawbacks of both model-based and data-driven methods of diagnosis.
The paper provides a new contribution where it is possible to create a hybrid FFD for industrial sensors. The contribution is clear and inedita. Some questions are indicated as showed bellow.
- In the first paragraph, industry 4.0 application examples can be supported by a relevant quote from the area where application examples are used.
- A sentence "three main categories; sensor faults, actuator and component/system faults", the authors describe different classes of faults. The sensor and actuator are ok, but about component/system as shown below It is needed to be revised. This type of fault, maybe, must be separated because It is possible having a fault in hardware systems such as a high temperature or low-efficiency processing of CPU. IN the other case, can be a fault due to bad implementation or coding the software.
- In "healthy state" it is not common in this case, could use "normal state".
- The final of section 2 where is presented a brief of related works it is suggested to insert a comparative table where the authors can light the real contribution of the proposed method, in order to show vantages and disadvantages.
- The sentence "In order to understand the mathematics behind RF, it is highly recommended to go through the explanation of decision trees and how they work in the first place." could be deleted.
- Equation of table 1 could be better presented as an equation in the text. What is the source citation of these equations?
- Algorithm 1 doesn't sound being an algorithm, but a description of steps. It is suggested change for algorithm steps or uses another way presentation.
- Section 4 is very short and doesn't bring important information such as (i) why is it used this source data from [27]? (ii) Is it possible to describe this machine plant test as a block diagram?
- Subsection 5.1 sounds to be part of the Material and Methods. Each block of figure 1 needs to be in deep described, especially where it is necessary to provide the type information used each one (input and output variables).
- At "The following table," the text could be removed.
- Based in Table 3, is it possible to affirm the method based on RF to improve techniques to identify faults, but no more than CART? Can the authors clarify what characteristics RF is better, such as, processing data information?
- Table 5 and also text present information about the accuracy of RF, but the accuracy is less than 1%. Is it correct? These values sounds are very low. 0.985% or 98.5%?
- In Figure 2 must be inserted x and y-label.
- Must insert the conclusion section at the end.
- At the Abstract and other parts of the manuscript, it is informed SQL queries was used. SQL is a language for database (DB) systems where is possible to create and select information from tables from DB, but It was not described the queries used such as "select from *". What kind of DB used?
Author Response
In the first paragraph, industry 4.0 application examples can be supported by a relevant quote from the area where application examples are used. A sentence "three main categories; sensor faults, actuator and component/system faults", the authors describe different classes of faults. The sensor and actuator are ok, but about component/system as shown below It is needed to be revised. This type of fault, maybe, must be separated because It is possible having a fault in hardware systems such as a high temperature or low-efficiency processing of CPU. IN the other case, can be a fault due to bad implementation or coding the software. In "healthy state" it is not common in this case, could use "normal state". The final of section 2 where is presented a brief of related works it is suggested to insert a comparative table where the authors can light the real contribution of the proposed method, in order to show vantages and disadvantages. The sentence "In order to understand the mathematics behind RF, it is highly recommended to go through the explanation of decision trees and how they work in the first place." could be deleted. Equation of table 1 could be better presented as an equation in the text. What is the source citation of these equations? Algorithm 1 doesn't sound being an algorithm, but a description of steps. It is suggested change for algorithm steps or uses another way presentation. Section 4 is very short and doesn't bring important information such as (i) why is it used this source data from [27]? (ii) Is it possible to describe this machine plant test as a block diagram? Subsection 5.1 sounds to be part of the Material and Methods. Each block of figure 1 needs to be in deep described, especially where it is necessary to provide the type information used each one (input and output variables). At "The following table," the text could be removed. Based in Table 3, is it possible to affirm the method based on RF to improve techniques to identify faults, but no more than CART? Can the authors clarify what characteristics RF is better, such as, processing data information? Table 5 and also text present information about the accuracy of RF, but the accuracy is less than 1%. Is it correct? These values sounds are very low. 0.985% or 98.5%? In Figure 2 must be inserted x and y-label. Must insert the conclusion section at the end. At the Abstract and other parts of the manuscript, it is informed SQL queries was used. SQL is a language for database (DB) systems where is possible to create and select information from tables from DB, but It was not described the queries used such as "select from *". What kind of DB used? Dear Dr. Silva, Thank you so much for accepting my invitation to review our work. Your valuable comments will be taken into consideration and attention. Below, I will reply to your comments one by one. Please not that your comments are the ones between quotations. Reviewer 1: "A sentence "three main categories; sensor faults, actuator and component/system faults", the authors describe different classes of faults. The sensor and actuator are ok, but about component/system as shown below It is needed to be revised. This type of fault, maybe, must be separated because It is possible having a fault in hardware systems such as a high temperature or low-efficiency processing of CPU. IN the other case, can be a fault due to bad implementation or coding the software." Author: Here the system described is an industry 4.0 mechanical or electronic. We are not talking about symptoms or warnings. Since human coding are counted as errors, not faults nor symptoms. As well as, any machine learning algorithm learns from the data provided by the system. So, we assume the data at least is accurate and coder or human inaccurate results to generate the data are not in our concern. Reviewer 1: "In "healthy state" it is not common in this case, could use "normal state"." Author: We are not using normal state. We are using the full efficiency state of all the system's components which is not normal, but the healthy state considered for this experiment. Reviewer 1: "The final of section 2 where is presented a brief of related works it is suggested to insert a comparative table where the authors can light the real contribution of the proposed method, in order to show vantages and disadvantages." Author: Thanks for the suggestion. It has been added in our research but was not included to the journal paper, since the scope was more applications that survey theory. Reviewer 1: "The sentence "In order to understand the mathematics behind RF, it is highly recommended to go through the explanation of decision trees and how they work in the first place." could be deleted." Author: In our research we have a detailed literature for decision tree and all the mentioned approaches. But it was not necessary for us to add them to the paper, since the journal is a technical journal and we assume that our target reader already knows. Reviewer 1: "Algorithm 1 doesn't sound being an algorithm, but a description of steps. It is suggested change for algorithm steps or uses another way presentation." Author: Thanks for the suggestion. The steps shows a Pseudo code of an algorithm such as random forests, hence it is an algorithmic representation of the method. Reviewer 1: "Section 4 is very short and doesn't bring important information such as (i) why is it used this source data from [27]? (ii) Is it possible to describe this machine plant test as a block diagram?" Author: Thank you for the suggestion. This article is a part of an ongoing research project and there the system and its components and research questions are answered. Besides this, the data was chosen because it is suitable to the research project's scopes and needed information. Reviewer 1: "Based in Table 3, is it possible to affirm the method based on RF to improve techniques to identify faults, but no more than CART? Can the authors clarify what characteristics RF is better, such as, processing data information?" Author: This observation has been already answered in the text right below the table. We have mentioned that we chose RF over CART because that RF is the optimization of CART to overcome overfitting problem. So CART might have a visible 0.4% higher accuracy (almost nothing) but it is highly likely that the results are overfitted to the training data, unlike RF. Reviewer1: "Table 5 and also text present information about the accuracy of RF, but the accuracy is less than 1%. Is it correct? These values sounds are very low. 0.985% or 98.5%?% Author: We have not claimed that the results are in Percentage. It is your personal assumption. the accuracy was 0.985 which is equivalent to 98.5% in percentage system. Which shows high accuracy for the mentioned algorithm. Reviewer 1: "Must insert the conclusion section at the end." Author: The conclusion is obviously been added. Please check the discussion and future work section. Reviewer 1: "At the Abstract and other parts of the manuscript, it is informed SQL queries was used. SQL is a language for database (DB) systems where is possible to create and select information from tables from DB, but It was not described the queries used such as "select from *". What kind of DB used?" Author: We have not claimed to use a database or SQL language in this work. It is left for future work. We show in this work a new architecture to use Random Forest as a hybrid approach between model-based and data driven approaches for diagnosis. If any other researcher or ourselves in the future would like to use this architecture for SQL rules or semantic rules such as SPARQL, it is completely possible. Thank you so much for the feedback. Best Regards, Ahlam MallakReviewer 2 Report
In this work, the architecture of a hybrid FDD method, between model-based and data-driven approaches to achieve FDD for component faults, is introduced. In this hybrid method, the datadriven part is represented by an optimized and hyperparameter tuned RF, in order to generate dynamic, diagnostic graphs that are later converted into a set of diagnostic rules and fed into a predefined system diagnostic model, acting as the model-based part of the proposed FDD system. The proposed approach provides a dynamic solution, unlike other model-based FDD approaches.
The work is technically sound and the paper is well written.
Author Response
In this work, the architecture of a hybrid FDD method, between model-based and data-driven approaches to achieve FDD for component faults, is introduced. In this hybrid method, the datadriven part is represented by an optimized and hyperparameter tuned RF, in order to generate dynamic, diagnostic graphs that are later converted into a set of diagnostic rules and fed into a predefined system diagnostic model, acting as the model-based part of the proposed FDD system. The proposed approach provides a dynamic solution, unlike other model-based FDD approaches. The work is technically sound and the paper is well written. Dear Reviewer 2, Thank you for the time and effort reviewing our paper, and for your positive feedback. It is such an honor. Wish you a pleasant day! Best Regards, Ahlam MallakReviewer 3 Report
This paper presents a hybrid method for sensor array fault diagnosis.
1.Industry 4.0 is an important development direction, which is worth studying. It is suggested to explain some content of Industry 4.0 in the Abstract, because industry 4.0 is mentioned in the Keywords section.
2.Can the author explain the difference between the fault diagnosis of traditional sensor system and the fault diagnosis of sensor system in Industry 4.0? Can the traditional sensor fault diagnosis method be applied to the fault diagnosis of sensor system in Industry 4.0?
3.The proposed method is consistent with the fault diagnosis method based on pattern recognition in essence.Data-driven methods can make data selection, such as PCA. The random forest method is equivalent to a classifier to realize the classification of features.What are the differences between the proposed method and pattern recognition?
4.The resolution of some pictures in this paper is low, and the space is too large, please adjust the author.For example, FIg.3 and FIg. 4.
Author Response
This paper presents a hybrid method for sensor array fault diagnosis. Dear Reviewer 3, Thank you so much for your time and effort in reviewing our work. I will reply to your comments one by one below. ^^ 1.Industry 4.0 is an important development direction, which is worth studying. It is suggested to explain some content of Industry 4.0 in the Abstract, because industry 4.0 is mentioned in the Keywords section. That is a great idea. The reader is expected to know industry 4.0 and its applications because this special issue of Sci Journal is addressed to take technologies in the field of industry 4.0. Thus, the reader is expected to be familiar with the topic. 2.Can the author explain the difference between the fault diagnosis of traditional sensor system and the fault diagnosis of sensor system in Industry 4.0? Can the traditional sensor fault diagnosis method be applied to the fault diagnosis of sensor system in Industry 4.0? Fault detection also is known as anomaly detection has well-defined branches and categories such as fault classification, clustering, encoder-decoder,..etc. You can definitely apply the traditional methods to detect faults in the industry 4.0 as any sensor system. However, keep in mind the inputs or modalities used to capture the data to train. If all the features are extracted from sensors, then it is going to be similar to traditional sensor systems. But, if there are other modalities like images, sound, signals..etc the process will differ. (at least the feature acquisition from the source data part will). 3.The proposed method is consistent with the fault diagnosis method based on pattern recognition in essence.Data-driven methods can make data selection, such as PCA. The random forest method is equivalent to a classifier to realize the classification of features.What are the differences between the proposed method and pattern recognition? Random forest (RF) is an optimization classifier (or regressor) for traditional classification and regression trees (CART). RF is a pattern recognition method, hence the fault detection with it belongs to the data-driven approaches. However, using the extracted rules by RF in a model-based schema can make RF a hybrid fault detection approach as explained in this method. -- Thank you! Best Regards, Ahlam MallakRound 2
Reviewer 1 Report
Based on the first revision.
Reviewer 1: "The final of section 2 where is presented a brief of related works it is suggested to insert a comparative table where the authors can light the real contribution of the proposed method, in order to show vantages and disadvantages." Author: Thanks for the suggestion. It has been added in our research but was not included to the journal paper, since the scope was more applications that survey theory.
Paper is based on the application, so a comparison can evidence the contribution of paper more than if was survey theory.
Reviewer 1: "Algorithm 1 doesn't sound being an algorithm, but a description of steps. It is suggested change for algorithm steps or uses another way presentation." Author: Thanks for the suggestion. The steps show a Pseudocode of an algorithm such as random forests, hence it is an algorithmic representation of the method.
The algorithm presented still the same. Including the previous comment about the shape it was done, the authors need to introduce all variables before the show the "algorithm", i.e., what is X, Y, x, y, S, f, fs, x, Ts. The same need to be done at Algorithm 2.
Reviewer1: "Table 5 and also text present information about the accuracy of RF, but the accuracy is less than 1%. Is it correct? These values sounds are very low. 0.985% or 98.5%?% Author: We have not claimed that the results are in Percentage. It is your personal assumption. the accuracy was 0.985 which is equivalent to 98.5% in percentage system. Which shows high accuracy for the mentioned algorithm.
But at the table, the column information is about "RF Accuracy (%)", so in this case, all information of its is 0.985% and also other numbers. Just verified the table.
Reviewer 1: "Must insert the conclusion section at the end." Author: The conclusion is obviously been added. Please check the discussion and future work section.
By the way, it is suggested changing for "Conclusion".
Regarding SQL language, the authors need to indicate this language only at the conclusion section as future works.
-------- Ending of questions about the reply of authors.
Regarding the rest of the manuscript.
Figure 4 has two parts, it is needed to separate them.
Figure 5 needs to be improved. For each tree, the sequence of parts is the same.
It is suggested authors provide a systematic revision about the comments of the first revision.
Author Response
It is suggested authors provide a systematic revision about the comments of the first revision. Dear Reviewer 1, Thank you so much for your detailed review and feedback. It was an honor to hear from you. Kindly check the replies related to your comments. The systematic revision requested is shown in details in the original.doc file we attached inside the .zip file. We highlighted each change and added comment notes. Paper is based on the application, so a comparison can evidence the contribution of paper more than if was survey theory. I believe the related work added to this work is sufficient. Since a full section that is called “Our contribution” is added previously and it shows what is our contribution to the state-of-the-art, then was followed by the related work section to prove it. The algorithm presented still the same. Including the previous comment about the shape it was done, the authors need to introduce all variables before the show the "algorithm", i.e., what is X, Y, x, y, S, f, fs, x, Ts. The same need to be done at Algorithm 2. The provided Pseudo codes previously named as “algorithm 1&2”. The algorithms had no problem since the beginning. All the parameters you mentioned are defined inside the algorithm. And there is no hard rule says that you have to declare all the variables in the beginning of the algorithm. However, to satisfy your request, we removed the words “Algorithm 1” and “Algorithm 2” from the pseudo codes if the word is what is causing the confusion. But at the table, the column information is about "RF Accuracy (%)", so in this case, all information of its is 0.985% and also other numbers. Just verified the table. It has been modified as requested. Thanks to you having a good eye and noticed it. We have removed (%) from the text and table 5. By the way, it is suggested changing for "Conclusion". The section name has been changed from “Discussion” to “Conclusion and Future work” based on your comment. Thank you very much! Best Regards, AuthorsReviewer 3 Report
The article meets the requirements for publication.
Author Response
The article meets the requirements for publication. Thank you so much! Best Wishes, Ahlam MallakRound 3
Reviewer 1 Report
Revisions, corrections, and updates were carried out as suggested by the reviewer.There is no reason to reject the manuscript.